# Novel Repositioning Therapy for Drug-Resistant Glioblastoma: In Vivo Validation Study of Clindamycin Treatment Targeting the mTOR Pathway and Combination Therapy with Temozolomide

**DOI:** 10.3390/cancers14030770

**Published:** 2022-02-02

**Authors:** Takeyoshi Eda, Masayasu Okada, Ryosuke Ogura, Yoshihiro Tsukamoto, Yu Kanemaru, Jun Watanabe, Jotaro On, Hiroshi Aoki, Makoto Oishi, Nobuyuki Takei, Yukihiko Fujii, Manabu Natsumeda

**Affiliations:** 1Division of Pharmacy, Medical and Dental Hospital, Niigata University, Niigata 951-8520, Japan; eda@med.niigata-u.ac.jp; 2Department of Neurosurgery, Brain Research Institute, Niigata University, Niigata 951-8585, Japan; masayasu_okd@bri.niigata-u.ac.jp (M.O.); oguryou@bri.niigata-u.ac.jp (R.O.); yoshi.tsukamoto@me.com (Y.T.); yu.k93@bri.niigata-u.ac.jp (Y.K.); watanabejun1003@yahoo.co.jp (J.W.); jotaro-on_silver@sky.hi-ho.ne.jp (J.O.); aoki1123@yahoo.co.jp (H.A.); mac.oishi@mac.com (M.O.); yfujii@bri.niigata-u.ac.jp (Y.F.); 3Department of Brain Tumor Biology, Brain Research Institute, Niigata University, Niigata 951-8585, Japan; nobtak0615@gmail.com

**Keywords:** glioblastoma, signal transduction, xenograft model, drug repositioning

## Abstract

**Simple Summary:**

Given the significant costs and lengthy timelines of drug development and clinical trials, drug repositioning is a promising alternative to find effective treatments for brain tumors quickly and inexpensively. In the present study, using a simple drug screen of macrolides, we found that clindamycin (CLD) had cytotoxic effects on glioblastoma (GBM) cells. Further studies showed the inhibition of the mammalian target of rapamycin (mTOR) pathway as the key mechanism of action. Interestingly, we found that co-treatment with temozolomide (TMZ), the alkylating agent considered as standard therapy in GBM, enhanced these effects and proposed the inhibition of O6-methylguanine-DNA methyltransferase (MGMT) protein by CLD as a potential mechanism for this combination effect.

**Abstract:**

Multimodal therapy including surgery, radiation treatment, and temozolomide (TMZ) is performed on glioblastoma (GBM). However, the prognosis is still poor and there is an urgent need to develop effective treatments to improve survival. Molecular biological analysis was conducted to examine the signal activation patterns in GBM specimens and remains an open problem. Advanced macrolides, such as azithromycin, reduce the phosphorylation of p70 ribosomal protein S6 kinase (p70S6K), a downstream mammalian target of rapamycin (mTOR) effector, and suppress the proliferation of T-cells. We focused on its unique profile and screened for the antitumor activity of approved macrolide antibiotics. Clindamycin (CLD) reduced the viability of GBM cells in vitro. We assessed the effects of the candidate macrolide on the mTOR pathway through Western blotting. CLD attenuated p70S6K phosphorylation in a dose-dependent manner. These effects on GBM cells were enhanced by co-treatment with TMZ. Furthermore, CLD inhibited the expression of the O6-methylguanine-DNA methyltransferase (MGMT) protein in cultured cells. In the mouse xenograft model, CLD and TMZ co-administration significantly suppressed the tumor growth and markedly decreased the number of Ki-67 (clone MIB-1)-positive cells within the tumor. These results suggest that CLD suppressed GBM cell growth by inhibiting mTOR signaling. Moreover, CLD and TMZ showed promising synergistic antitumor activity.

## 1. Introduction

Glioblastoma (GBM) accounts for the majority of primary brain tumors and is considered as a grade IV glioma based on the WHO classification system [1]. The median survival of patients with GBM is approximately 15 months [2,3]. Despite the development of multimodal therapy with surgical resection, radiation, and temozolomide (TMZ)-based chemotherapy, GBM prognosis remains poor. GBM is characterized by diverse genetic and epigenetic alterations, such as isocitrate dehydrogenase (IDH) 1/2 mutations, O6-methylguanine-DNA methyltransferase (MGMT) promoter methylation, epidermal growth factor receptor (EGFR) amplification, and EGFR variant III (EGFRvIII) expression [4,5,6,7]. In particular, EGFR amplification, PTEN mutations, CDKN2A deficiency, and TP53 gene mutations were investigated to establish the association between genetic alterations and GBM prognosis [8]. Comprehensive genomic analysis revealed aberrant signal transduction through various pathways, including the receptor tyrosine kinase (RTK)–Ras-phosphoinositide 3-kinase (PI3K) pathway, Rb pathway, and p53 pathway [9,10]. These studies suggest that genetic alterations or RTK–Ras–PI3K, Rb, and p53 pathways act cooperatively and contribute to the proliferation and maintenance of GBM cells [11]. Further studies are required to better understand the GBM etiology and to develop novel molecular targeted strategies.

Macrolide antibiotics (macrolides) act on Gram-positive bacteria by reversibly binding to the bacterial 50S ribosomal subunit and inhibiting peptide elongation. Accumulating evidence showed that macrolides have immunomodulatory properties in vitro [12]. Macrolides exhibit atypical pharmacological profiles and inhibit the synthesis and secretion of pro-inflammatory cytokines, including tumor necrosis factor (TNF)-alpha, interleukin (IL)-1, IL-6, and IL-8 [13,14]. Long-term or low-dose administration of macrolides reduces the symptoms of diffuse panbronchiolitis [15]. A previous study showed that cell proliferation and cytokine secretion by CD4+ T-cells were inhibited by azithromycin treatment [16]. In addition, azithromycin suppressed mammalian target of rapamycin (mTOR), which is frequently activated in cancer cells [16,17,18,19].

Herein, we focused on the atypical features of macrolides and examined the possibility of applying these drugs to conventional chemotherapy for GBM. During our primary assessment, we investigated the effects of various macrolide compounds, namely, azithromycin, clarithromycin, clindamycin (CLD), and erythromycin, on the growth and survival of human GBM cell lines and found that CLD remarkably suppressed GBM cell proliferation. However, our understanding of the molecular mechanism underlying CLD-mediated cytotoxicity is incomplete. We hypothesized that the mTOR signaling pathway might be a potential target of CLD. To verify this, we investigated the effects of CLD on growth and survival in several GBM cell lines and explored the specific molecular targets of CLD with a focus on p70S6K, a substrate of mTOR. We further tested the efficacy of CLD in vivo, either alone or in combination with TMZ, using the mouse xenograft model.

## 2. Materials and Methods

### 2.1. Reagents

For the in vitro assays, TMZ (TEMODAR capsules; MSD, Japan) and clindamycin hydrochloride (Dalacin capsules; Pfizer, Japan) were used. TMZ was dissolved in dimethyl sulfoxide (DMSO) and stored at −20 °C after preparing a 100 mM stock solution. CLD was dissolved in sterile water to produce a 330 mM stock solution and stored at −20 °C. For the in vivo treatment, TMZ (TEMODAR injection; MSD, Tokyo, Japan) and CLD (clindamycin phosphate; Dalacin S injection; Pfizer, Tokyo, Japan) were administered to the animals after diluting them in an appropriate solvent.

### 2.2. Cell Culture

Cells were grown in DMEM containing 10% fetal bovine serum (Sigma-Aldrich, St. Louis, MO, USA). Human glioblastoma cell lines (U251, T98G, and LN229) were obtained from the American Type Culture Collection (ATCC). NGT41 is a cell line established from a disseminated lesion of the cervical spinal cord from a BRAF V600E-mutant epithelioid glioblastoma patient and cultured as previously reported [20].

### 2.3. Cell Viability Assays

The cytotoxic effects of TMZ and CLD were determined by using the cell viability assay reagent WST-1 (Takara Bio, Japan), as described previously [21]. Briefly, glioma cells were seeded at the density of 1–1.5 × 10^3^ cells/well in 96-well flat-bottomed plates and incubated at 37 °C overnight. Afterward, the cells were treated with TMZ (0, 62.5, 125, 250, or 500 μM) or CLD (0, 110, 220, 440, or 660 μM) for 72 h. Following treatment, WST-1 reagent (10 μL) was added to each well and incubated for 1–2 h at 37 °C. Absorbance was measured at 450 nm using a microplate reader. The viability of untreated cells was considered as 100%.

### 2.4. Western Blotting

Cells or tumor tissues were lysed and sonicated in lysis buffer (50 mM Tris-HCl, pH 7.5, 150 mM NaCl, 2% sodium dodecyl sulfate (SDS), 10 mM NaF, 2 mM Na_3_VO_4_, 5 mM EDTA 1 mM phenylmethylsulfonyl fluoride) and CompleteTM Protease inhibitor cocktail (Roche, Indianapolis, IN, USA). The lysates were centrifuged and the supernatant was collected. The protein concentration in the lysates was determined using the Micro BCA Protein assay kit (Thermo Fisher Scientific, Rockford, MI, USA). Equal amounts of protein (15–30 μg) were subjected to SDS–polyacrylamide gel electrophoresis (PAGE) and transferred to a nitrocellulose membrane. Western blotting was performed using anti-phospho-p70S6K (Thr389) (1:500), anti-phospho-S6 (Ser240/244) (1:1000), anti-S6 (1:4000), anti-phospho-4EBP1 (Thr37/46) (1:1000), and anti-β-actin (1:4000) polyclonal antibodies (Cell Signaling Technologies, Danvers, MA, USA). Anti-p70S6K antibody was purchased from Santa Cruz Biotechnology (Santa Cruz, CA, USA). Membranes were incubated with the indicated primary antibodies at 4 °C overnight. After the membranes were rinsed with TBST (50 mM Tris-HCl pH 7.5 and 150 mM NaCl containing 0.1% Tween 20) and subsequently incubated with horseradish-peroxidase-conjugated secondary antibodies (1:10000; Santa Cruz CA, USA). Immunoreactivity was detected via the chemiluminescence detection method using the ECL system (Bio-Rad, Hercules, CA, USA). The immunoreactive bands were visualized using GeneGnome (Syngene, Cambridge, UK) and quantified using Genetools software (Syngene). The dose dependence of CLD on p70S6K phosphorylation at Thr389 was examined in cultured cells after 72 h of treatment. The levels of phosphorylated and total p70S6K were analyzed via Western blotting and quantified using the Syngene Bio Imaging system (Syngene, Cambridge, UK). The phospho/total p70S6K ratio was determined. β-actin was used as the loading control. For the original Western blots, see the Appendix A.

### 2.5. In Vitro mTOR Kinase Assays

The immunoprecipitation and kinase assays were performed as described previously [22,23]. Briefly, cells were lysed in ice-cold buffer A (50 mM Tris-HCl pH 8.0, 150 mM NaCl, 1 mM EGTA, 5 mM EDTA, 20 mM glycerophosphate, 0.5 mM dithiothreitol, CompleteTM, and PhosStopTM phosphatase inhibitor cocktails (Roche, San Francisco, CA, USA)). The supernatants from the centrifuged samples were incubated with Protein G SepharoseTM (GE Healthcare, Chicago, IL, USA), coupled with anti-mTOR (N5D11) antibody (IBL, Fujioka, Japan) for 2 h at 4 °C. The immunocomplex was washed and the kinase assay was initiated by adding the reaction buffer (10 mM HEPES, 50 mM glycerophosphate, 50 mM NaCl, 10 mM MgCl_2_, 4 mM MnCl_2_, 250 μM ATP, and recombinant GST-4EBP1 (1 μg/sample)). After incubation for 20 min at 30 °C, the reaction was terminated by adding the SDS sample buffer. The samples were boiled and subjected to SDS-PAGE. Western blotting was performed with anti-phospho-4EBP1 (Thr37/46) antibody to detect the mTOR complex (mTORC) 1 activity.

### 2.6. Cell Cycle Analysis

Tumor cells treated with CLD (440 μM) for 24 h were trypsinized and washed with PBS. At least 1 × 10^5^ cells were fixed with ice-cold 70% ethanol at 4 °C. The cells were collected via centrifugation and stained with propidium iodide (PI) using the MUSE cell cycle reagents (Millipore, Billerica, MA, USA) following the manufacturer’s instructions, as described previously [24].

### 2.7. Experimental Animals

Four-week-old male nude mice (BALB/C-nu/nu, Charles River Laboratories Inc., Yokohama Japan) were used for the in vivo experiments. Mice were housed under aseptic conditions in a plastic cage and provided free access to food and water. Each cage was kept in a colony room (22 ± 1.0 °C) under a 12 h light–dark cycle. All of the animal experiments described here were approved by the Animal Committee of Niigata University (No. SA00519) and were performed in accordance with the Guiding Principles for the Care and Use of Laboratory Animals (NIH, Bethesda, MD, USA).

### 2.8. Establishment of a Patient-Derived Xenograft Model

For the subcutaneous tumor model, NGT41 cells were suspended in Neurobasal Medium (Life Technologies Corporation, Carlsbad, NY, USA) and implanted subcutaneously (1 × 10^6^ cells per place) into the nude mice as described previously [20]. When the tumor volume reached 50 mm^3^, the mice were randomly divided into four groups. Mice were treated daily with solvent (vehicle control), TMZ (5 mg/kg), CLD (400 mg/kg), and TMZ (5 mg/kg) + CLD (400 mg/kg) via intraperitoneal injection for 10 days. The tumor size was measured daily with calipers, and tumor volume was calculated using the formula: tumor volume (mm^3^) = [length (mm) × width (mm^2^)]/2 [25].

### 2.9. Histopathological Examination

Mice were deeply anesthetized with medetomidine hydrochloride (0.3 mg/kg), midazolam (4 mg/kg), and butorphanol tartrate (5 mg/kg) via intraperitoneal injection, and tumors were resected. The tumors were fixed with 4% neutral-buffered paraformaldehyde, dehydrated using 70% ethanol, and embedded in paraffin wax. Sections (3 μm thick) were cut from the paraffin blocks, and immunostaining was performed as described previously [26]. Sections were stained with 3,3′-diamonobenzidine as a chromogen and counterstained with hematoxylin solution (Wako Chemicals Inc., Osaka, Japan), and then processed for the HE staining and immunostaining. The following mouse monoclonal antibodies were used: anti-Ki-67 monoclonal antibody (clone MIB-1; 1:100; Biogenex, Fremont, CA, USA) and anti-MGMT monoclonal antibody (clone MT3.1; 1:50; Chemicon International, Temecula, CA, USA; 1:50), MGMT immunoreactivity was evaluated in representative areas of the tumors showing the characteristic features [27].

### 2.10. Statistical Analyses

Data were expressed as mean ± SE and were subjected to parametric analyses. When univariate data with more than two groups showed a similar distribution, we used the analysis of variance (ANOVA) to assess the statistical differences in terms of time, dose, and experimental groups, followed by Bonferroni’s post hoc test for multiple comparisons. Alternatively, Student’s *t*-test (two-tailed) was used for univariate data analysis of two groups. All statistical analyses were performed using the GraphPad Prism 6 software (GraphPad Software, San Diego, CA, USA).

## 3. Results

### 3.1. Antiproliferative Activity of Clindamycin in Cultured Glioblastoma Cells

Macrolides are known to regulate inflammation and immune responses. We investigated the effects of approved macrolides on cell growth using human GBM cell lines. In our preliminary experiments, we treated U251 and T98G cells with azithromycin, clarithromycin, clindamycin (CLD), and erythromycin. The results showed that CLD inhibited the growth and survival of the U251 and T98G cells (Appendix A). Moreover, CLD also inhibited the growth of the U251, T98G, LN229, and NGT41 cell lines (Figure 1A–D). The inhibitory effect of CLD on these cells was dose-dependent. The minimal dose of CLD used to reach a 50% inhibitory effect was observed at 440–660 μM (U251, 440 μM; T98G, 500 μM; LN229, 660 μM; and NGT41, 500 μM). Thus, a dosage of 440 μM was used for subsequent experiments. These results suggested that CLD suppressed the growth of malignant glioma cells in vitro.

### 3.2. Effects of Clindamycin on mTOR Signaling in Glioblastoma Cell Lines

To investigate the effects of the CLD on mTOR signaling, we examined p70S6K phosphorylation via Western blotting using the anti-phospho p70S6K (Thr389) antibody. Results showed that CLD treatment reduced the p70S6K phosphorylation in a dose-dependent (Figure 2A,C,E,G) and time-dependent manner (Figure 3A–H). CLD also inhibited the phosphorylation of ribosomal S6 protein (S6), a substrate of p70S6K, in a dose-dependent manner (Figure 2B,D,F,H), indicating that CLD inhibited the phosphorylation of p70S6K and S6 proteins in a dose-dependent manner.

### 3.3. Direct Effect of Clindamycin on mTOR Kinase Activity

To verify the inhibitory effect of CLD on mTOR signaling, we measured the mTOR kinase activity in vitro. Cells were treated with CLD (440 μM) for 72 h and then lysed to collect the protein. Equal amounts of protein from treated or untreated cells were subjected to immunoprecipitation with anti-mTOR antibody (N5D11). Using recombinant GST-4EBP1 as a substrate, the kinase activity of mTORC1 was evaluated. The results showed that CLD-treated cells had lower mTOR kinase activity compared to untreated cells (Figure 4A,B), suggesting that CLD directly inactivated mTOR signaling in GBM cells.

### 3.4. Effect of Clindamycin on Cell Cycle Progression in GBM Cells

To investigate the effects of CLD on the cell cycle progression, we treated NGT41 and T98G cells with 440 μM CLD for 24 h and performed cell cycle analyses. The results showed that treatment with CLD induced G0/G1 phase arrest (Figure 4C). The percentage of NGT41 cells in the G0/G1 phase increased from 56.7% to 75.8% following CLD treatment (Figure 4D). In contrast, CLD treatment decreased the percentage of cells in the S-phase (from 14.5% to 7.3%) and G2/M phase (from 25.4% to 14.3%) of the cell cycle. Similar but more modest effects were observed in T98G cells (Appendix A). Further, we examined the expression of the cell-cycle-related protein cyclin D1 via Western blotting. The results showed that treatment with CLD reduced the expression of cyclin D1 (Figure 4E).

### 3.5. Effects of Clindamycin and Temozolomide Combined Treatment on Glioma Cell Viability

Next, we investigated the effects of CLD and TMZ combination treatment on human glioblastoma cell lines. Each cell line was treated with a vehicle, CLD, TMZ, or CLD in combination with TMZ for 72 h. In our preliminary experiments, we found that the minimum concentration of TMZ that was not totally cytotoxic and could reduce cell viability was 125 μM (Appendix A). Co-treatment with CLD and TMZ significantly increased the cytotoxicity of MGMT-positive cell lines, T98G, and NGT41 (Figure 5A,B). The same additive effect of CLD was not observed in MGMT-negative cell lines, U251 and LN229 (Figure 5C,D). To elucidate the mechanism underlying the effect of combined CLD and TMZ treatment, we examined the MGMT protein level via Western blotting. The results showed that CLD suppressed MGMT in a dose-dependent manner in T98G and NGT41 cell lines (Figure 5E,F). MGMT mRNA expression was not affected by CLD in both cell lines (Appendix A).

### 3.6. Synergistic Effects of Clindamycin and Temozolomide in a Subcutaneous Tumor Model

Having confirmed the antitumor activity of the CLD and TMZ combined treatment in NGT41 cells, we next aimed to evaluate the combinatorial effects of these drugs in vivo using an orthotopic xenograft model. Nude mice bearing NGT41 xenografts were administered a vehicle, CLD, TMZ, or CLD in combination with TMZ daily for 10 days. The results showed that the combination treatment with CLD + TMZ significantly suppressed the tumor growth in vivo (Figure 6A). In contrast, CLD monotherapy had no effect on tumor growth in NGT41 xenografts. There were no adverse effects, such as weight loss or gastrointestinal dysfunction in animals that received the combination treatment (Appendix A). At the end of the treatment (day 10), the subcutaneous tumors were resected. Histopathological examination of these tumors showed that the combined treatment markedly decreased the number of MIB1-positive cells (Figure 6C). Moreover, Western blotting showed that the combined treatment suppressed the expression of MGMT in tumors, confirming its therapeutic potential during long-term treatment in vivo (Figure 6B).

## 4. Discussion

GBM is one of the most aggressive tumors of the central nervous system (CNS). Despite multiple preclinical studies and the development of molecular targeted therapies, the overall survival of patients with GBM has not improved significantly. GBM has diverse genetic alterations and spatial and temporal heterogeneity, which have been implicated in chemotherapy resistance. In this study, we examined the anti-proliferative effects of the approved macrolides in human GBM cell lines. CLD markedly reduced the viability of GBM cells and inhibited p70S6K phosphorylation. Since PI3K–Akt–mTOR axis abnormalities are implicated in the etiology of GBM and other types of brain tumors [19,28,29,30], the activation of the mTOR pathway in the CNS is a subject of intense research. mTOR is a serine/threonine kinase that forms mTOR complex (mTORC) 1 and 2, depending on the binding partners. mTORC1 is activated by amino acids and growth factors and plays a central role in cell growth and proliferation. Upon mTORC1 activation, the phosphorylation of its substrates, such as p70S6K and eukaryotic initiation factor 4E binding protein (4EBP), is increased. Thus, the phosphorylation ratio of these molecules is often used as an index of mTORC1 activity [31,32]. Consistent with our hypothesis, CLD attenuated the phosphorylation of p70S6K and S6 ribosomal protein in a dose-dependent manner. In vitro kinase assays revealed that CLD directly affected the mTORC1 activity. These results suggested that CLD inhibited the mTOR signaling that is required for GBM growth.

To delineate the mechanism underlying the anti-proliferative effect of CLD, we performed cell cycle analysis. Treatment with CLD increased the percentage of cells in the G0/G1 phase and decreased the percentage of cells in the S and G2/M phases, suggesting that CLD induced G0/G1 arrest. There are several reports showing that mTOR inhibitors suppress cell proliferation. Rapamycin treatment or mTOR depletion induces cell cycle arrest in the G1/S phase and leads to a decrease in nucleolar size [33]. Rapamycin and its derivatives, namely, CCI-779 and RAD001, inhibit the phosphorylation of p70S6K and 4EBP1, leading to G1-phase cell cycle arrest [34]. It might be possible that aberrant PI3K–Akt–mTOR signaling contributes to proliferation in gliomagenesis, and the cytotoxic effects of CLD are mediated via mTOR.

Interestingly, our results showed that in MGMT-positive cell lines, namely, T98G and NGT41, the antitumor effect of CLD was significantly enhanced by TMZ co-treatment. Moreover, CLD suppressed the levels of MGMT protein in these cells, and long-term treatment with CLD and TMZ decreased the MGMT in xenografts. Furthermore, CLD and TMZ combination therapy markedly reduced the number of MIB1-positive cells within the tumor. This suggests that CLD suppresses MGMT protein expression and enhances the antitumor activity of TMZ. However, the long-term administration of CLD alone failed to suppress tumor growth. It was reported that after mTOR inhibition, pathway signaling plasticity with redundant signal input and reactivation by a feedback loop occurs [35]. Our results are consistent with the previous report. Therefore, combinatorial approaches for manipulating the PI3K–Akt axis and inhibiting the mTOR kinase to inactivate the feedback loop have been proposed [36]. Preclinical studies demonstrated the efficacy of XL765, a PI3K/mTOR dual inhibitor, in combination with TMZ against GBM xenografts [37,38,39]. Results of the present study suggested that CLD was effective in combination with TMZ.

In TMZ-based chemotherapy, where DNA mismatch and subsequent apoptosis occurs, resistance has been associated with MGMT expression [40]. Dose-dense TMZ or TMZ and interferon combination therapy was proposed as a potential treatment strategy to deplete MGMT, although all large clinical trials attempting to exploit this have failed [41,42]. We observed the effectiveness of CLD and TMZ combination in MGMT-positive cell lines. CLD may sensitize these cells to TMZ by reducing the MGMT protein level. CLD did not affect the MGMT mRNA, thus CLD suppressed MGMT protein during the translation processes or at the protein degradation levels. Since mTORC1 controls translation and autophagy [43], CLD decreases MGMT protein possibly through these mechanisms.

CLD exerts an antibacterial effect by binding to the 50S ribosomal subunit of susceptible bacteria, causing a reduction in the cessation of protein synthesis; however, its effects on eukaryotes are not well understood. Thus, the exact molecular mechanism by which CLD reduces MGMT protein remains unknown and further investigation is warranted.

We investigated the effect of CLD on NGT41 cell proliferation. NGT41 is a cell line that was established from an epithelioid GBM harboring the BRAF V600E mutation. We previously confirmed the NGT41 tumor cell response to combination therapy with dabrafenib and trametinib in vitro and in vivo [20]. mTOR pathway activation is thought to be one of the main mechanisms of resistance in BRAF V600E-mutant brain tumors following targeted treatment [44,45,46]. This further supports the notion of integration of the Ras–ERK and PI3K–mTORC1 pathways and the mechanisms of crosstalk include negative feedback loops, cross-inhibition, cross-activation, and pathway convergence on substrates [47]. mTORC1 is the center of the crosstalk, as it receives several inputs from both Ras–ERK and PI3K signaling pathways. Our findings suggest that mTOR inhibition by CLD is effective in treating BRAF V600E mutant glioblastomas.

Our cell culture experiments revealed that treatment with 110–440 μM (i.e., 50–200 μg/mL) CLD significantly attenuated the phosphorylation of p70S6K and S6K in GBM cell lines. However, in our animal studies, we did not determine the concentration of CLD or its metabolites in the serum following CLD injection. It is important to consider whether these ranges of CLD can be achieved in the CNS. In adult humans, the peak concentration of CLD in the plasma following 900 mg intravenous injection of CLD phosphate is approximately 14.1 μg/mL. For severe infections, intravenous administration of 1200–4800 mg per day has been used (Pfizer Inc., medical information; CLEOCIN phosphate injection, http://www.pfizer.com, accessed on 1 December 2021) Thus, the CLD dose used in our culture experiments was higher than that used in a clinical setting. Imaoka et al. reported that in adult rats, a single intravenous injection of CLD (30 mg/kg) resulted in a peak concentration of 48 μg/mL in the lungs; however, the distribution of CLD in the brain was found to be relatively lower [48]. CLD cannot cross the blood–brain barrier, and it is known that CLD does not reach the cerebrospinal fluid. In our experiments, repeated and high-dose (400 mg/kg) administration had no adverse effect, such as significant weight loss on the mice (Appendix A). In a mouse endotoxic shock model, pretreatment with CLD (440 mg/kg, intraperitoneal) suppressed the release of inflammatory cytokines [49]. The effective concentration of CLD used in the present study appeared to be tolerable in mice. Further research is required to understand how CLD and its combination treatment have a local impact on CNS regarding the proliferation and activation of GBM cell metabolism. To improve drug distribution to the brain, methods such as convection-enhanced delivery [50,51] and intranasal delivery [52,53] of nanoliposomal drugs, use of nanoparticles [54], and blood–brain barrier disruption by focused ultrasound [55] were investigated and could be applied to CNS delivery of CLD.

Drug repositioning, or identifying and developing new uses of existing drugs, is a promising method used in the pharmaceutical industry to reduce the cost of developing new therapeutical drugs [56]. Our group [25], as well as others [57], performed drug repositioning to identify candidate drugs for use against glioblastomas and other brain tumors [58]. Specifically, antibiotics have gained considerable interest in drug repurposing for cancer therapy [59]. In the present study, through a simple screening of four macrolides, we found that the commonly used CLD exerts cytotoxic effects on GBM cell lines and is a candidate for drug repositioning.

## 5. Conclusions

In conclusion, in GBM cell lines, CLD inhibited the mTOR signaling, which is frequently dysregulated in cancers. Moreover, CLD enhanced the susceptibility of GBM cells toward TMZ-based therapy, which is presently a standard treatment for GBM. We speculate that a CLD and TMZ combination therapy prevented the development of drug resistance by reducing MGMT through mTOR signal downregulation. CLD is an exciting candidate drug for repositioning, and combined CLD and TMZ therapy is a potentially new and useful strategy for targeting glioblastomas.

## Figures and Tables

**Figure 1 cancers-14-00770-f001:**
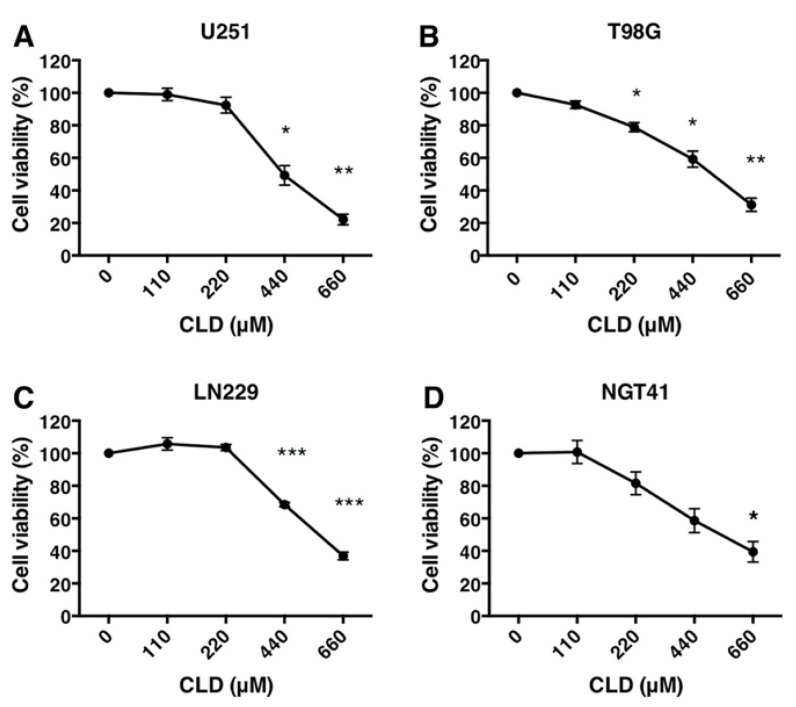
Cytotoxic effects of clindamycin on cultured glioblastoma cell lines. Dose responses of clindamycin (CLD) on cell viability in glioblastoma cell lines (**A**–**D**). Cells were treated with CLD (0, 110, 220, 440, or 660 μM) for 72 h and subjected to a WST-1 cell viability assay. The viability of untreated cells (vehicle control: 0 μM CLD) was considered 100%. Data are presented as the mean ± SE. Similar results were obtained from triplicate experiments (* *p* < 0.05, ** *p* < 0.01, and *** *p* < 0.001 vs. control culture: one-way ANOVA).

**Figure 2 cancers-14-00770-f002:**
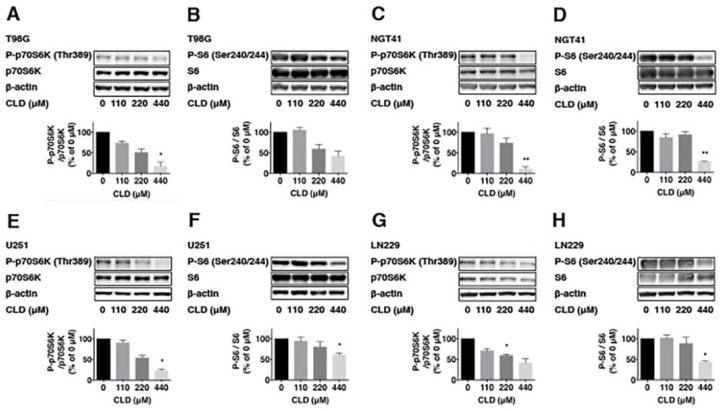
Effect of clindamycin on the phosphorylation of p70S6K and S6 in glioblastoma cell lines. Dose responses of clindamycin (CLD) on the phosphorylation of p70S6K at Thr389 and S6 at Ser240/Ser244 in glioblastoma cell lines. Cells were treated with CLD (0, 110, 220, or 440 μM) for 72 h and subjected to Western blotting. The levels of phosphorylated and total p70S6K were indicated at the corresponding bands (**A**,**C**,**E**,**G**). Western blotting analysis was subjected to densitometric quantification after standardizing the ratio of phospho/total p70S6K (below panel). The phosphorylation and total S6 levels in glioblastoma cell lines were similarly analyzed via Western blotting (**B**,**D**,**F**,**H**). β-actin was used as the internal control level in all cases. Data are shown as the mean ± SE, similar results shown were representative of three independent experiments. Each group was compared to the 0 concentration level to determine statistical significance. (* *p* < 0.05 and ** *p* < 0.01 vs. control culture: one-way ANOVA).

**Figure 3 cancers-14-00770-f003:**
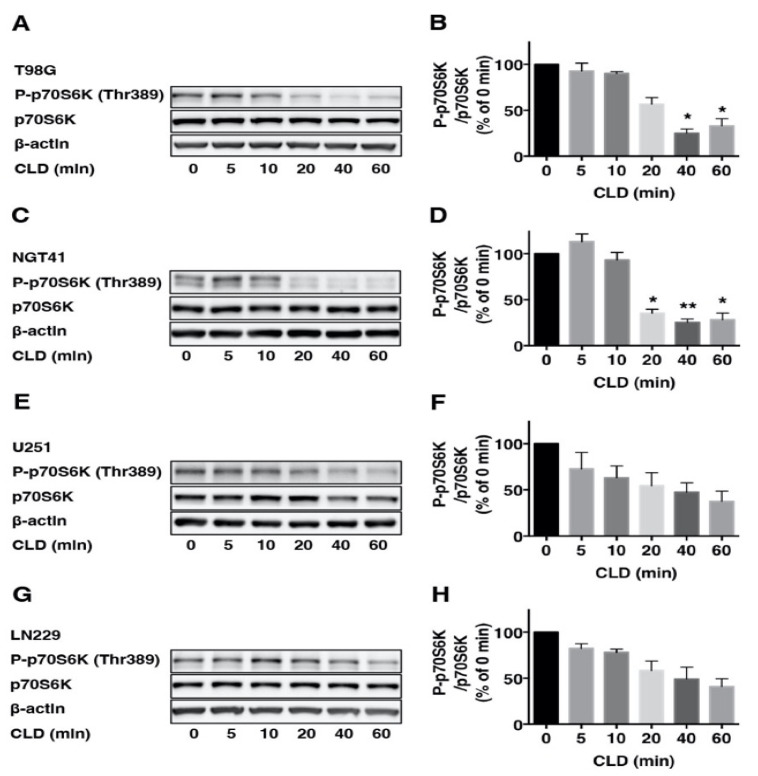
Time course effect of clindamycin on the phosphorylation of p70S6K in glioblastoma cell lines. Time course analysis of clindamycin (CLD) on p70S6K phosphorylation was determined in glioblastoma cell lines (**A**,**C**,**E**,**G**). Cells were assigned to the indicated times and cultured with CLD (440 μM) and subjected to Western blotting. The levels of phosphorylation and total p70S6K were quantified after standardizing the ratio of phospho/total p70S6K (**B**,**D**,**F**,**H**). β-actin was used as the internal control level in all cases. Data are shown as the mean ± SE, similar results were obtained from three independent experiments. Each group was compared to the 0 min level to determine the statistical significance (* *p* < 0.05 and ** *p* < 0.01 vs. control culture: one-way ANOVA).

**Figure 4 cancers-14-00770-f004:**
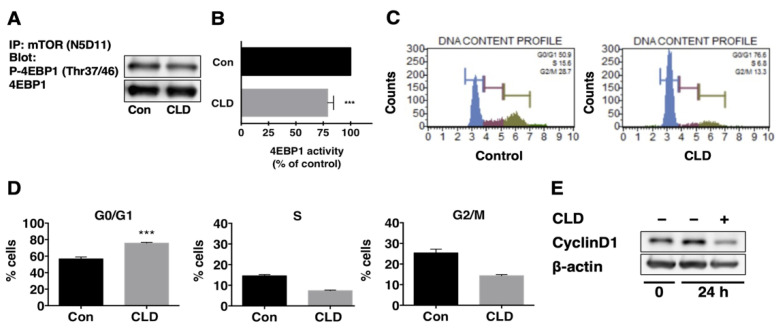
Direct effect of clindamycin on mTORC1 kinase activity and its involvement in the cell cycle progression at NGT41 cells. The in vitro kinase assay was examined in NGT41 cells. Cells were treated with a vehicle (control) or clindamycin (CLD) 440 μM for 72 h. Cell lysates were immunoprecipitated with the anti-mTOR (N5D11) antibody. The immunoprecipitates were added 4EBP1 as a substrate and were subjected to SDS-PAGE. mTORC1 activity was evaluated via Western blotting using anti-phospho 4EBP1 at the Thr37/46 antibody (**A**). The intensity of the immunoreactive band was quantified after standardizing the ratio of phospho/total 4EBP1 (**B**). Data are shown as the mean ± SE, where results were obtained from three independent experiments. *** *p* < 0.001 vs. control culture: *t*-test. NGT41 cells were treated with CLD (440 μM) for 24 h and subjected to cell cycle analysis (**C**). Cells were classified into three phases: G0/G1, S, and G2/M (**D**). Data represent the percentage of cells in each phase, revealed as mean ± SE; similar results were obtained from three independent experiments. *** *p* < 0.005 vs. control: *t*-test. The expression of cyclin D1 was analyzed via Western blotting (**E**). β-actin was used as the loading control.

**Figure 5 cancers-14-00770-f005:**
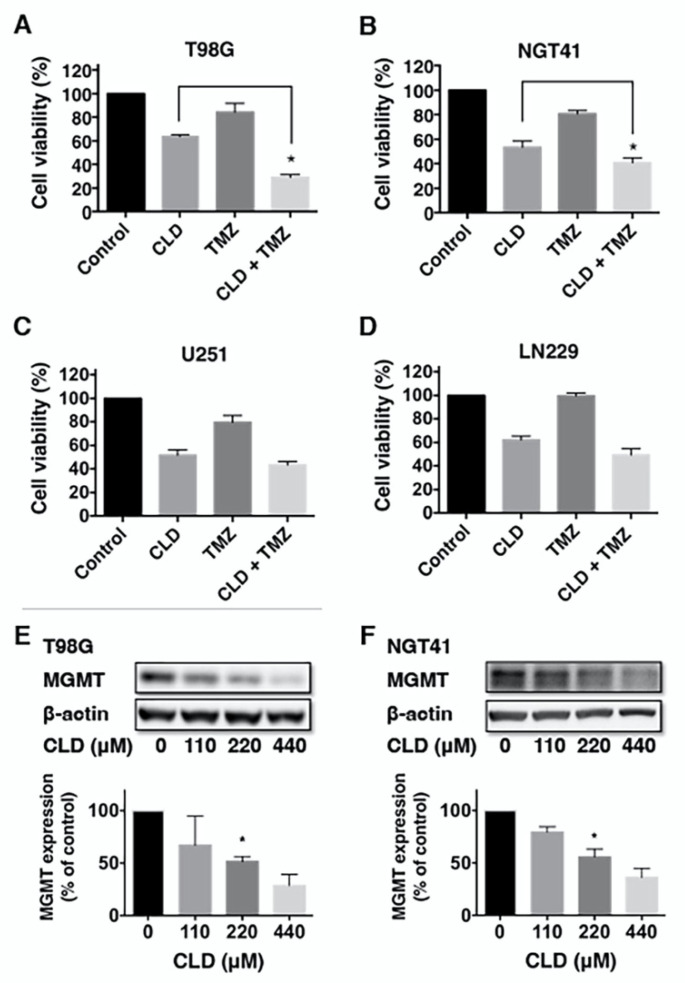
Effect of clindamycin and temozolomide combined treatment on glioblastoma cell viability. Combination effect of clindamycin (CLD) and temozolomide (TMZ) on glioblastoma cell viability (**A**–**D**). Cells were treated with a vehicle control, CLD (440 μM), TMZ (125 μM), or CLD (440 μM) in combination with TMZ (125 μM) for 72 h and subjected to a WST-1 cell viability assay. Data are presented as a percentage ratio to the vehicle control and are shown as the mean ± SE; similar results were representative of five independent experiments. The *p*-values were generated using a post hoc test for multiple comparisons, comparing CLD + TMZ with CLD alone in each cell line (* *p* < 0.05: one-way ANOVA). The dose response of CLD (0, 110, 220, or 440 μM) on the MGMT protein level was examined in T98G and NGT41 cell lines. After treatment for 72 h, cell lysates were subjected to Western blotting using an anti-MGMT antibody (**E**,**F**). β-actin was used as the loading control.

**Figure 6 cancers-14-00770-f006:**
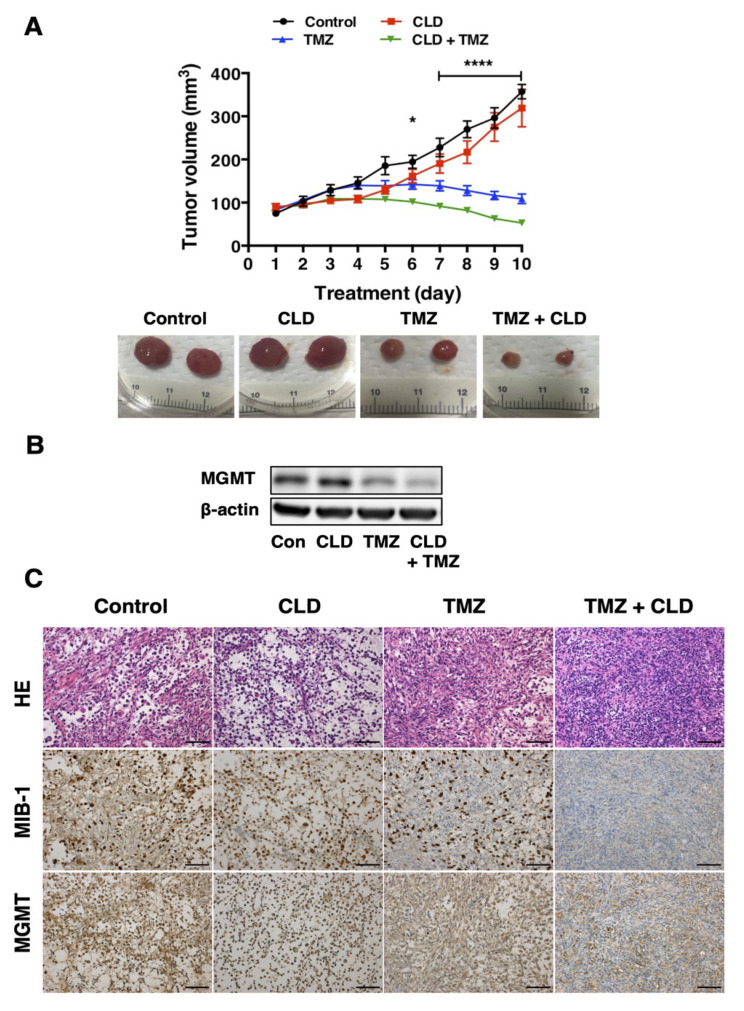
Synergistic effect of the clindamycin and temozolomide combined treatment in a subcutaneous tumor model. Efficacy of combined treatment for NGT41 subcutaneous xenograft model. Data represent the relationship between treatment days and tumor volume (**A**). Mice were administered a vehicle control, CLD (400 mg/kg), TMZ (5 mg/kg), or CLD (400 mg/kg) in combination with TMZ (5 mg/kg) for 10 days. Data are shown as means ± SE (*N* = 7). The *p*-values represent statistical comparisons between the CLD alone and combined treatment groups using Bonferroni multiple comparisons between independent samples (* *p* = 0.0299 and **** *p* < 0.0001 vs. CLD alone: two-way ANOVA). Subcutaneous tumors were resected on day 10 and then homogenized. The lysates were subjected to Western blot analysis using an anti-MGMT antibody (**B**). The subcutaneous tumor resected from the NGT41 xenograft and its histological images are shown (**C**). Representative micrographs of subcutaneous tumors using HE and immunohistochemical staining with anti-MIB-1 and anti-MGMT antibodies on day 10. The results shown are representative of four independent experiments. Scale bar: 100 μm.

## Data Availability

The datasets analyzed during the current study are available from the corresponding author upon request.

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
