# Peer review of "Novel Repositioning Therapy for Drug-Resistant Glioblastoma: In Vivo Validation Study of Clindamycin Treatment Targeting the mTOR Pathway and Combination Therapy with Temozolomide"

_cancers, 2022, doi:10.3390/cancers14030770_

Round 1

Reviewer 1 Report

The paper is well written and figures are very good.  Only question I have is the use of cell viability as equal to cell proliferation as in Fig. 5?

Author Response

The paper is well written and figures are very good.  Only question I have is the use of cell viability as equal to cell proliferation as in Fig. 5?

We thank the reviewer for an important comment. We completely agree with the reviewer’s comment that cell proliferation and cell viability are not complete equal; proliferation and apoptosis/cell death should be evaluated to further understand the results of cell proliferation. We have now changed the wording when describing the results of cell viability assays as not to use the term “proliferation”.

We have looked at the proliferation marker MIB-1 (equal to Ki67) in our in vivo studies and found that combined clindamycin and temozolomide markedly reduces proliferation (Fig 6).

Reviewer 2 Report

Drug repositioning is an important subject for cancer science, especially for aggressive brain tumours as GBM. In this paper, Takeyoshi Eda at al demonstrate that clindamycin in combination with temozolomide represents a possible therapy for GBM acting on the mTOR pathway. As a general comment the paper is scientifically valid, well structured and written.

Here below some concerns:

line 188, Fig S1: please indicate the doses of the different drugs used in the experiment

line 190-192: the 50% inhibitory effect of CLD indicated for the different cell lines are not the real calculated calculated, instead authors indicate the minimal dose used to reach a 50% inhibitory effect. Please modify the statement

Fig 2 legend: reference to panels are confusing and misleading.  Reference to panels A, C, E and G should be put after p70S6K at Thr389 and reference to panels B, d, F and H should be put after S6 at Ser240/Ser244

Fig 2: Please check the statistical significance of 440 uM CLD in lower panels of the histograms in panels B and G

Paragraph 3.5: Please give a comment in the results or in the discussion section about the absence of the effect described in U251 and LN229 cell lines

Fig 6A: a statistical analysis of the effects between TMZ and DLD+TMZ would be appreciated. Moreover the upper row of panel C should be linked to panel A an not to panel C

Paragraph 3.6: There is no reference for Fig 6A

Discussion: given the difficulty in reaching the CNS, authors are invited to speculate about new mechanisms of drug delivery (liposomes, nanovesicles?)

Author Response

Reply to reviewer 2

Drug repositioning is an important subject for cancer science, especially for aggressive brain tumours as GBM. In this paper, Takeyoshi Eda at al demonstrate that clindamycin in combination with temozolomide represents a possible therapy for GBM acting on the mTOR pathway. As a general comment the paper is scientifically valid, well structured and written.

Here below some concerns:

line 188, Fig S1: please indicate the doses of the different drugs used in the experiment.

We thank the reviewer for the suggestion. We have added the dose of each drug (all 100 µg/mL) below the x-axis and figure legend.

line 190-192: the 50% inhibitory effect of CLD indicated for the different cell lines are not the real calculated calculated, instead authors indicate the minimal dose used to reach a 50% inhibitory effect. Please modify the statement.

We thank the reviewer for a keen observation. We have not calculated the real IC50 in each cell line, but indicated the minimal dose to reach a 50% inhibitory effect We have now corrected the wording according to your suggestion (lines 150-151).

Fig 2 legend: reference to panels are confusing and misleading.  Reference to panels A, C, E and G should be put after p70S6K at Thr389 and reference to panels B, d, F and H should be put after S6 at Ser240/Ser244

We apologize for the confusing figure legend. We have corrected the figure legend for Figure 2 according to your indication.

Fig 2: Please check the statistical significance of 440 uM CLD in lower panels of the histograms in panels B and G

We thank the reviewer for thoroughly looking at out data. We have double checked our statistical analysis. We used ANOVA to assess differences in more than two groups with Gaussian distribution, followed by Bonferroni’s multiple comparison test to assess to compare the mean of each column with those of other columns. P-values were insignificant for 440 µM CLD compared with control in Fig 2B (p = 0.1778 by Tukey’s comparison) and Fig 2G (p = 0.2298, p = 0.0729 by Tukey’s correction ), probably due to slightly high error.

Paragraph 3.5: Please give a comment in the results or in the discussion section about the absence of the effect described in U251 and LN229 cell lines

We have now added the absence of significant additive effect of CLD in U251 and LN229 (lines 242-243).

Fig 6A: a statistical analysis of the effects between TMZ and CLD+TMZ would be appreciated. Moreover, the upper row of panel C should be linked to panel A an not to panel C

We have now corrected figure 6 according to the reviewer’s suggestion. Statistical comparison of TMZ and CLD + TMZ groups were performed, but statistically significant differences were not detected by Bonferroni’s comparison test between the two groups, p = 0.3468 (day 6), p = 0.1550 (day 7), p = 0.1845 (day 8), p = 0.688 (day 9), p = 0.0516 (day 10).

Paragraph 3.6: There is no reference for Fig 6A

We thank the reviewer for pointing out this error. We have now referenced Fig 6A in the manuscript (Lines 300-301).

Discussion: given the difficulty in reaching the CNS, authors are invited to speculate about new mechanisms of drug delivery (liposomes, nanovesicles?)

We thank the reviewer for suggesting an important point. As the reviewer points out, improving CNS penetration may further increase efficacy of CLD in vivo, as CLD has proven difficult to reach to CNS. We have now added a section briefly reviewing new methods of drug delivery which could potentially be used for delivering CLD (lines 407-410).